# Geospatial distribution and predictors of postnatal care utilization during the critical time in Ethiopia using EDHS 2019: A spatial and geographical weighted regression analysis

**Muluken Chanie Agimas** \*, **Tigabu Kidie Tesfie, Nebiyu Mekonnen Derseh, Meron Asmamaw, Habtamu Wagnew Abuhay, Getaneh Awoke Yismaw**

Department of Epidemiology and Biostatistics, Institute of Public Health, College of Medicine, University of Gondar, Gondar, Ethiopia

\* mulukensrc12@gmail.com

## Abstract

### Introduction

Postnatal care within 2 days after delivery is classified as early postnatal care. Maternal and neonate mortality during the early postnatal period is a global health problem. Sub-Saharan Africa contributes the highest maternal and newborn mortality rates. To reverse this problem, early postnatal care is the best strategy, but there is no study to show the spatial distribution and application of geographical weighted regression to show the effect of each predictor on early postnatal care across the geographic areas in Ethiopia using the recent EDHS 2019 data.

### Objective

To assess the geospatial distribution and predictors of postnatal care utilization during the critical period in Ethiopia using EDHS 2019.

### Method

A secondary data analysis of a cross-sectional study was used among 2105 women. The data for this analysis was taken from the 2019 EDHS, and missing data was managed by imputation. The spatial variation of postnatal care during the critical time was assessed using the Getis-Ord Gi* statistic; Moran's I statistics were conducted to test the autocorrelation; and Sat Scan statistics were also used to show the statistically significant clusters of early PNC utilization in Ethiopia. The ordinary least squares method was used to select factors explaining the geographical variation of postnatal care during the critical time. Finally, the geographical weighted regression was used to show the spatial variation of the association between predictors and outcomes. Predictors at 95% CI with a p-value <0.05 were statistically significant factors for PNC during the critical time.

**Data Availability Statement:** All relevant data is within the manuscript

**Funding:** The author(s) received no specific funding for this work.

**Competing interests:** The authors have declared that no competing interests exist.

**Abbreviations:** AIC, Akaike information criteria; ANC, Antenatal Care; DHS, Demographic Health Survey; EDHS, Ethiopian Demographic Health Survey; GWR, Geographic Weighed Regression; LLR, Log Likelihood Ratio; OLS, Ordinary Least Square; PNC, Postnatal Care; RR, Relative Risk.

## Results

The overall prevalence of PNC utilization during critical time was 713 (34%, 95%CI: 31.5%–36.5%). The spatial distribution of postnatal care utilization during critical times was not randomly distributed across the area of Ethiopia. The hotspot areas of postnatal care utilization during the critical period in Ethiopia were found to be in Benishangul, Gumuz, and the western part of Tigray. Whereas, the cold spot area was in the western part of the southern nation and nationality of Ethiopia. Women with antenatal care visits, facility delivery, no education, and media exposure were the predictors of postnatal care utilization during the critical time in the hotspot areas of Ethiopia.

## Conclusion and recommendation

In Ethiopia, one-third of women utilize the PNC during critical times. Postnatal care utilization during critical times was not randomly distributed across the regions of Ethiopia. Antenatal care visits, facility delivery, lack of education, and media exposure were the predictors of postnatal care utilization during the critical time in Ethiopia. Therefore, encouraging facility delivery, awareness creation by expanding media access, and literacy are highly recommended to improve the utilization of PNC services during this critical time in Ethiopia.

## Introduction

Postnatal care is the most important health service provided for the mother and newborns immediately after birth, up to 42 days [1, 2]. According to the recommendations of the World Health Organization, regardless of the place where birth occurs, women should attend the PNC service within 24 hours after birth, within 48–72 hours, and between 7–14 days and 6 weeks after birth to improve maternal and newborn health [3]. Postnatal care (PNC) within 2 days after delivery is classified as early PNC [4]. Early PNC is responsible for the high rate of maternal and child deaths [5]. PNC utilization is a very important health outcome for both maternal and child survival, and the majority of maternal and neonatal mortality can be reduced by timely PNC services [6]. According to the previous report, about 88% to 98% of all maternal deaths can be reduced by PNC service [7]. In Ethiopia, just using PNC utilization, newborn deaths can be reduced by 10–27% [5]. But according to a global report, within 24 hours of birth, about 4 million neonates and 287,000 maternal deaths occurred each year, and the majority of deaths occurred in low-income countries, including Ethiopia [8]. The maternal mortality ratio and neonatal mortality are higher in Ethiopia, which accounts for about 412 per 1,000 live births and 30 per 1,000 live births, respectively [9]. In Ethiopia, 50% and 40% of all maternal and neonatal deaths occur within 24 hours [5]. The majority of deaths are overcome through immediate PNC service [8–11]. Evidence showed that PNC utilization was significantly associated with institutional delivery utilization, place of residence, ethnicity, pregnancy intention, antenatal care visit, place of delivery, marital status, wealth quintile, maternal education, education of partner, skilled delivery, and age [8–10]. The majority of the previous studies were focused on factors associated with PNC in general, whereas studies on PNC utilization during the critical period were not adequate. Even though one national-level study was conducted on early PNC in Ethiopia using EDHS 2016 data and another local-level study was conducted in Sidama, Ethiopia [12, 13], it did not allow us to know about the effects of each predictor across the geographical areas of Ethiopia. Because all possible factors are not

equally statistically significant for PNC utilization during the critical time across the regions. Additionally, the spatial distribution of PNC utilization during the critical time is not equally distributed across the corner of Ethiopia. So the spatial distribution of PNC utilization during the critical time is highly important to policymakers, planners, and other stakeholders for evidence based practices. As far as we know, there is no study about the spatial distribution and application of geographical weighted regression to show the effect of each predictor across the geographic areas of Ethiopia. Therefore, the current study aimed to assess the geospatial distribution and predictors of postnatal care utilization during the critical time in Ethiopia using EDHS 2019 using a spatial and geographical weighted regression analysis.

## Methods

### Study design and setting

The cross-sectional study design was conducted in Ethiopia using the EDHS-2019 data set. Ethiopia is located in east Africa at 30–150 N latitude and 330–480 E longitude [14]. Ethiopia occupies about 1.1 million square kilometres, and its latitude ranges from 148 meters below sea level (Dallol, Afar, Ethiopia) to the maximum peak at Ras Dashen (Amhara, Ethiopia), with an above-sea level of 4,620 meters [5] (**Fig 1**).

### Population

All women who gave birth in the previous 2 years were the source population, and all women who gave birth in the previous 2 years in the enumeration area were the source population.

### Eligibility criteria

**Inclusion criteria.**

- Women whose ages 15–49 who gave birth prior to the 2019 EDHS survey were either permanent residents or visitors who slept in the household the night before the survey were eligible to be interviewed.

- This study included women who had given birth within 2 years preceding the survey and had at least one postnatal check, whether before discharge from a health facility after birth or after home delivery or discharge from the health facility.

**Exclusion criteria.**

- Women who were seriously ill during the data collection period were excluded.

### Variables

The dependent variable for this study was PNC utilization during the critical time (yes/no) and the independent variables are shown in (**Table 1**).

### Possible bias in the measurement of PNC during the critical time

Recall and interviewer bias may affect the accuracy of the classification.

### Sampling procedure and sampling technique

The EDHS 2019 of Ethiopia was used for the current spatial and geographical weighted regression and two-stage sampling, such as cluster selection and selection of the enumeration areas (EAs). The second step was the systematic sampling of households in the selected EAs, leading

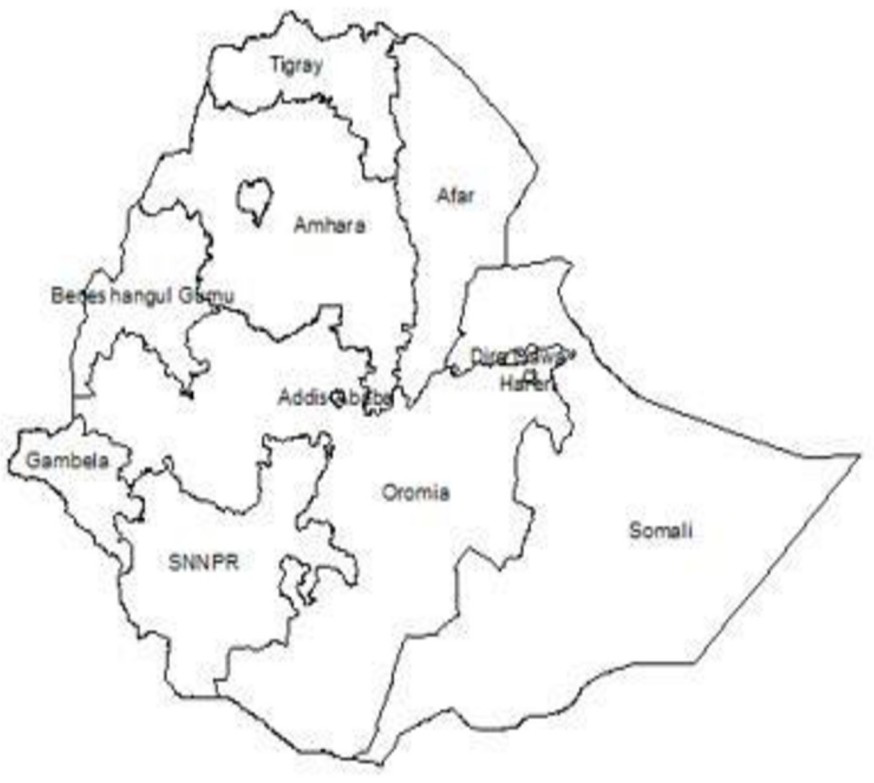

Source of the map: https://africaopendata.org/dataset/ethiopia-shapefiles

**Fig 1. Map of Ethiopia.**

to a total of 9160 households. The data used for the current analysis was from households and women's questionnaires. From 9160 households, there were 8,855 reproductive women ages 15–49. Among these 8,855 women, 2105 women (a weighted sample) had given birth within 2 years preceding the survey and had at least one postnatal check, whether before discharge from a health facility after birth or after home delivery or discharge from the health facility. These 2105 women were the total sample size (486 from urban areas and 1619 from rural areas) for the current study and were selected by the multistage sampling method. We used this sampling technique more than other techniques because of the wide geographical area. Other sampling methods can be used. The detailed sampling method and procedure were reported in the 2019 EDHS report [15].

## Data source

The data for this analysis was taken from the 2019 EDHS.

## Data collection procedures and data quality assurance

Initially, the current utilized data was collected during EDHS 2019, and for the purpose of the current secondary data analysis, such data was requested online from the International Demographic Health Survey (DHS) and accessed on the DHS's official website link at, https://www.

**Table 1. Study variables for the study of postnatal care utilization during the critical time hot spots in Ethiopia using EDHS 2019.**

| Variable | Description |
|---|---|
| **Dependent variable** | |
| PNC utilization during the critical time (early PNC) | In this study, it refers to mother who have at least one postnatal visit within 48 hours following delivery [4].<br>**Critical time:** the time in which maternal and new-born deaths are high, or the time of the first two days after delivery. |
| **Independent variables** | |
| wealth index | The wealth index is defined by the wealth of the participant using principal component analyses and factor analyses, which are classified as poor, middle, and rich. |
| Educational status | The educational achievement of the participant can be classified as no education, primary education, secondary education, or above educational level. |
| Place of delivery | Defined by the actual living place of the participants, either urban or rural. |
| Religion | The religion of the participant was assessed by the structured response options as Orthodox Christian, Muslim, protestant, Catholic, traditional religion, and others. |
| cesarean delivery | Women who delivered surgically. If she has a history of surgical delivery labelled as yes and otherwise classified as no. |
| Media exposure: | Was assessed based on whether people had access to read newsletters, listen to the radio, and watch TV. Accordingly, if they have access to all three media (newsletter, radio, and TV) at least once a week, we categorized them as "yes", otherwise "no" [14]. |
| Age of the women | The age of the participant was collected in years (numerically). Then the numerical value was categorized as <20 years, 20–34 years, and 35–49 years. |
| ANC visit | Mothers were asked about the history of ANC service utilization, which has been classified as either 'yes' or 'no visit'. |
| Place of delivery | We asked women about where they give birth and labelled them as "at home" or "at health institutions." |
| current pregnant | Women were asked about the status of their pregnancy and classified as "yes" or "no". |
| Region | The region is the administrative structure or boundary of Ethiopia, which is Amhara, Tigray, Oromo, Afar, Somalia, Harari, Benishangul Gumuz, Gambela, the south nation and nationality of people, and two self-governed cities such as Diredawa and Addis Ababa. |
| Birth order | Women were asked about their birth order and classified as "1," "2–3," "4–5," and "≥6." |

dhsprogram.com/data/dataset/Ethiopia_Interim-DHS_2019.cfm?flag=1. The data was also collected through a pretested, structured interview-administered method. Training for field workers and pretests were conducted to assure the quality of the data. The geographic coordinates of each cluster or the location data were collected by the Global Positioning System receivers. The displacement of urban clusters was 2 kilometres and 5 kilometres for rural clusters, and to assure the confidentiality of the participants' information, the geographic positioning system's (GPS) latitude and longitude of all clusters were random. The detailed part about data collection and quality assurance was reported in EDHS 2019 [15].

## Data management and analysis

After the data was accessed, the dependent and independent variables were extracted, cleaned, and recoded for further analysis. The frequency of both the dependent and independent variables was also weighted using the sampling weight technique before analysis. For descriptive

analysis, STATA software version 17 was used. After weighting the data, the data was exported to ArcGIS version 10.3 for spatial and geographical weighted regression analysis.

## Spatial analysis

**Autocorrelation.** Using ArcGIS version 10.3, Moran's I statistics were conducted to test the autocorrelation. We used the global Moran's I statistics to show a holistic characterization, providing a single value for each region. Moran's I statistics range from 1 to -1. Moran's I value close to -1 indicated that the distribution of early PNC is dispersed across the regions of Ethiopia, and its value close to +1 indicated that the distribution of early PNC is clustered, whereas Moran's I value of zero showed that early PNC is distributed randomly. Moran's p-value less than 0.05 was the cutoff point for the significance of spatial autocorrelation [16].

**Hot/cold spot analysis.** The Getis-Ord statistic was also used to show the degree of clustering or to show the hot or cold spot area of early PNC utilization in Ethiopia.

**Spatial interpolation.** Showing the early PNC utilization of all parts of Ethiopia is difficult. Therefore, based on the sampled data or area, predicting the non-sampled area is a critical issue. To achieve this objective, the ordinary Kriging spatial interpolation technique was used to predict the non-sampled area. Closer things are more related than distant ones, and so the spatial dependence of early PNC utilization was also considered. We used this method because it has the advantages of being the best linear unbiased estimator and because it accounts for clustering and screening effects.

**Sat scan analysis.** Sat Scan statistics using Kuldorff Sat Scan version 9.6 software were also used to show the statistically significant clusters of early PNC utilization in Ethiopia [17]. The maximum cluster size per population was 50%. Spatial incremental autocorrelation was also assessed to estimate the appropriate distance threshold of the spatial process, which promotes clustering to calculate an appropriate.

## Spatial regression

**Ordinary least squares regression.** After identifying the hot spot of PNC utilization during the critical time, the observed spatial patterns of the predictors of early PNC were identified by spatial regression modeling. The trustworthiness of ordinary least squares regression (OLS) depends on the fluffiness of all the necessary assumptions [18]. In the OLS regression model, the coefficient of each independent variable should be significantly associated with either a protective or risk factor for the outcome variable. The OLS model also should be free from Multicollinearity, non-stationary, the residual must not have a spatial pattern, a model should have key predictors, and the residual must be free from spatial autocorrelation [18–20]. The Multicollinearity between independent variables was assessed by the variance inflation factor (VIF). Independent variables with a VIF less than 7.5 are declared to have no Multicollinearity.

The data mining tool and explanatory regression were used to select a model that fulfilled the assumptions of OLS regression. The explanatory regression can select those models with a better adjusted $R^2$ score and select a model that satisfies all the OLS regression assumptions [18]. The final model was validated by internal cross-validation. For a better prediction of the model, use a mean error close to zero, a small root-mean-square error, and an average standard error. The root-mean-squared standardized error must be approaching one [21]. Our model satisfies all the former assumptions.

**Geographically weighted regression.** The effect of each predictor may vary across the clusters. This non-stationary characteristic can be assessed by using geographically weighted regression (GWR). The single linear regression equation for the study area is possible using

GWR but not using OLS. For each cluster, the equation can be formulated using GWR. But OLS uses the data from all clusters. Therefore, for each cluster, different GWR coefficients can be considered. The GWR model [22] can be written as:

$$Yi = \beta0(uivi) + \sum_{k=1}^{p} \beta k(uivi)xik + \varepsilon i$$

Where:
Yi = observation of response
(*uivi*) = latitude and longitude
*βk* (*uivi*) (*k* = 0, 1, . . . *p*,) are *p* unknown functions of geographic locations (*uivi*).
*xik = independent variables* at location (*uivi*), *i* = 1,2,. . .n
*εi* = residuals with zero mean and homogeneous variance σ$^2$.

## Ethical declaration

Since it was a secondary data analysis of EDHS, informed consent from the participants was not applicable. Rather, data requests and approval for access were obtained from DHS International. All data was fully anonymized before we accessed informed consent from DHS International.

## Results

A total of 2105 participants were included in the analysis. About 1051 (49.9%) gave birth at a health facility, and about 1271 (60.4%) of the participants had a history of ANC visits. Furthermore, 1673 (79.5%) of them had media exposure (**Table 2**).

### Prevalence of PNC utilization during critical time

The overall prevalence of PNC utilization during critical times in Ethiopia was 34% (95% CI: 31.5%–36.5%).

### Spatial analysis

**Spatial autocorrelation.** The spatial distribution of PNC utilization during the critical time in Ethiopia was none-randomly distributed across the clusters, with Moran's I statistics of 0.027 (p = 0.023) (**Fig 2**).

### Spatial distribution and hot spot analysis

A total of 305 clusters were used for the spatial distribution of PNC utilization during the critical time in Ethiopia. Thus, the highest proportion of PNC utilization during the critical time in Ethiopia was in the western part of Amhara (**Fig 3B**). Additionally, the hotspot area of PNC utilization during the critical time in Ethiopia was found to be in Benishangul Gumuz and the western part of Tigray (**Fig 3A**). Whereas, the cold spot area was in the western part of the southern nation and nationality of Ethiopia. For more detail, see **Fig 3A**.

### Incremental autocorrelation

The number of bands for the incremental autocorrelation was ten. At a distance of 150 kilometers (km) with a significant z-score value, the spatial clustering PNC utilization during the critical time was highly pronounced (**Fig 4**).

**Table 2. Characteristics of the women aged 15–49 years who give birth in the 2 years preceding to 2019 EDHS in Ethiopia.**

| Variable | Category | Weighted frequency | % |
|---|---|---|---|
| Wealth index | Poor | 1547 | 73.5 |
| | Middle | 392 | 18.6 |
| | Rich | 166 | 7.9 |
| Education status | No education | 1305 | 62 |
| | Primary education | 512 | 24.3 |
| | Secondary and above | 288 | 13.7 |
| Place of delivery | Home | 1054 | 50.1 |
| | Health facility | 1051 | 49.9 |
| Religion | Orthodox Christian | 590 | 28.02 |
| | Muslim | 1088 | 51.69 |
| | Catholic | 12 | 0.56 |
| | Protestant | 385 | 18.3 |
| | Traditional | 23 | 1.08 |
| | Other | 7 | 0.35 |
| Cesarean delivery | Yes | 128 | 6.1 |
| | No | 1977 | 93.9 |
| Residence | Urban | 486 | 23.08 |
| | Rural | 1619 | 76.92 |
| Age of women | <20 | 621 | 29.5 |
| | 20–34 | 720 | 34.2 |
| | 35–49 | 764 | 36.3 |
| ANC visit | Yes | 1271 | 60.4 |
| | No | 834 | 39.6 |
| Birth order | 1 | 669 | 31.8 |
| | 2–3 | 792 | 37.6 |
| | 4–5 | 381 | 18.1 |
| | ≥6 | 263 | 12.5 |
| media exposure | Yes | 1673 | 79.5 |
| | No | 432 | 20.5 |
| Current pregnant | Yes | 438 | 20.8 |
| | No | 1667 | 79.2 |
| Region | Amhara | 185 | 8.8 |
| | Oromia | 263 | 12.5 |
| | Somali | 232 | 11 |
| | Afar | 240 | 11.4 |
| | SNNPR | 242 | 11.5 |
| | Gambela | 166 | 7.9 |
| | Benishangul Gumuz | 196 | 9.3 |
| | Tigray | 166 | 7.9 |
| | Harari | 162 | 7.7 |
| | Addis Ababa | 105 | 5 |
| | Dire Dawa | 148 | 7 |

**Note**, others = none believer

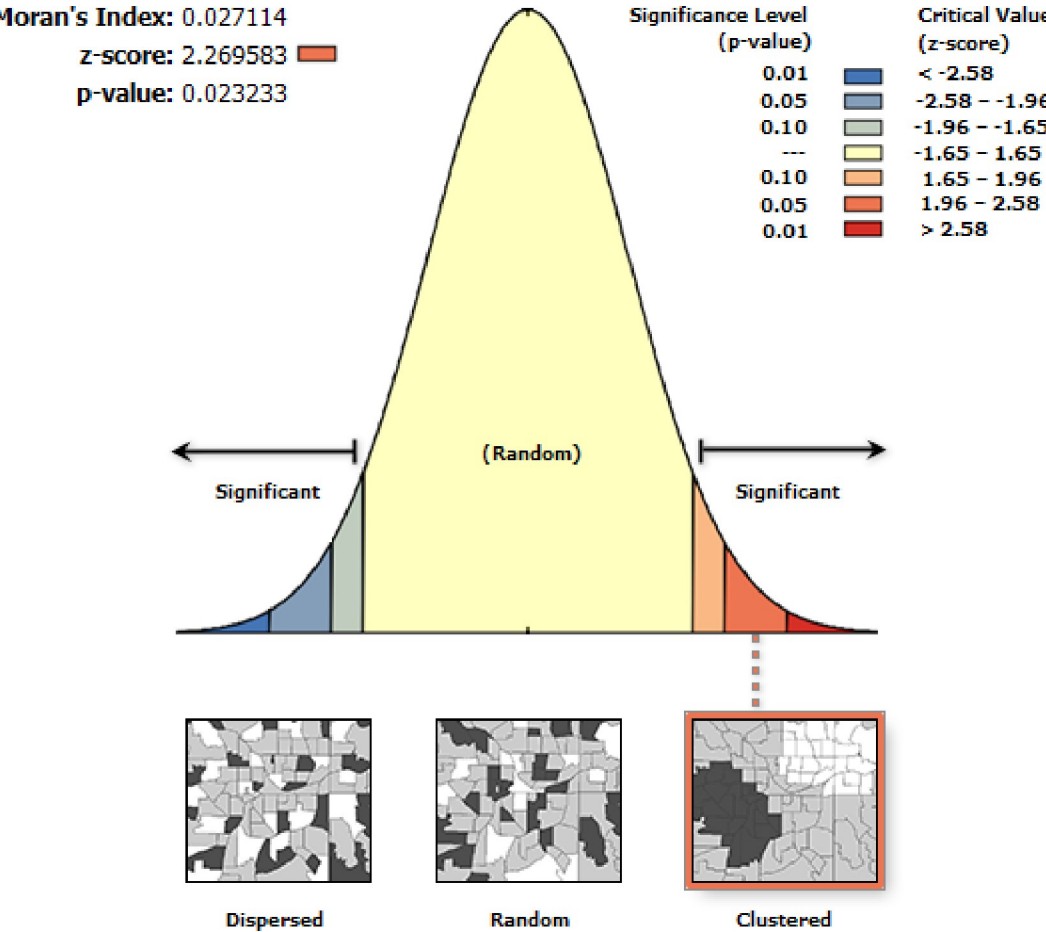

**Fig 2. Spatial Autocorrelation of PNC utilization during the critical time in Ethiopia using EDHS-2019.**

### Spatial interpolation

As the spatial ordinary kriging method showed, the highly predicted utilization of PNC during the critical time in Ethiopia was in the Benishangul Gumuz region. Whereas, the lower predicted utilization of PNC during the critical time was in the Somalia region, the south-eastern part of the Oromia region, some parts of the eastern part of the Amhara region, and the southern Afar region (**Fig 5A**).

### SaT scan analysis

Out of a total of 305 clusters, 28 were statistically significant clusters. As presented in Fig 5B, the red window represents the primary clusters of PNC utilization during critical times in Ethiopia. Of the total statistically significant clusters, 8 clusters were primary (most likely) cluster types; the rest 20 clusters were secondary, tertiary, and quarterly and fifthly clusters. The primary cluster was located at 10.196300 N and 34.606731 E within a 41.99 km radius in Benishangul Gumuz. Furthermore, the PNC utilization during critical times in Ethiopia was 3.69 times higher than outside the window (RR = 3.69, LLR = 28.77, p-value < 0.001) (**Table 3 and Fig 5B**).

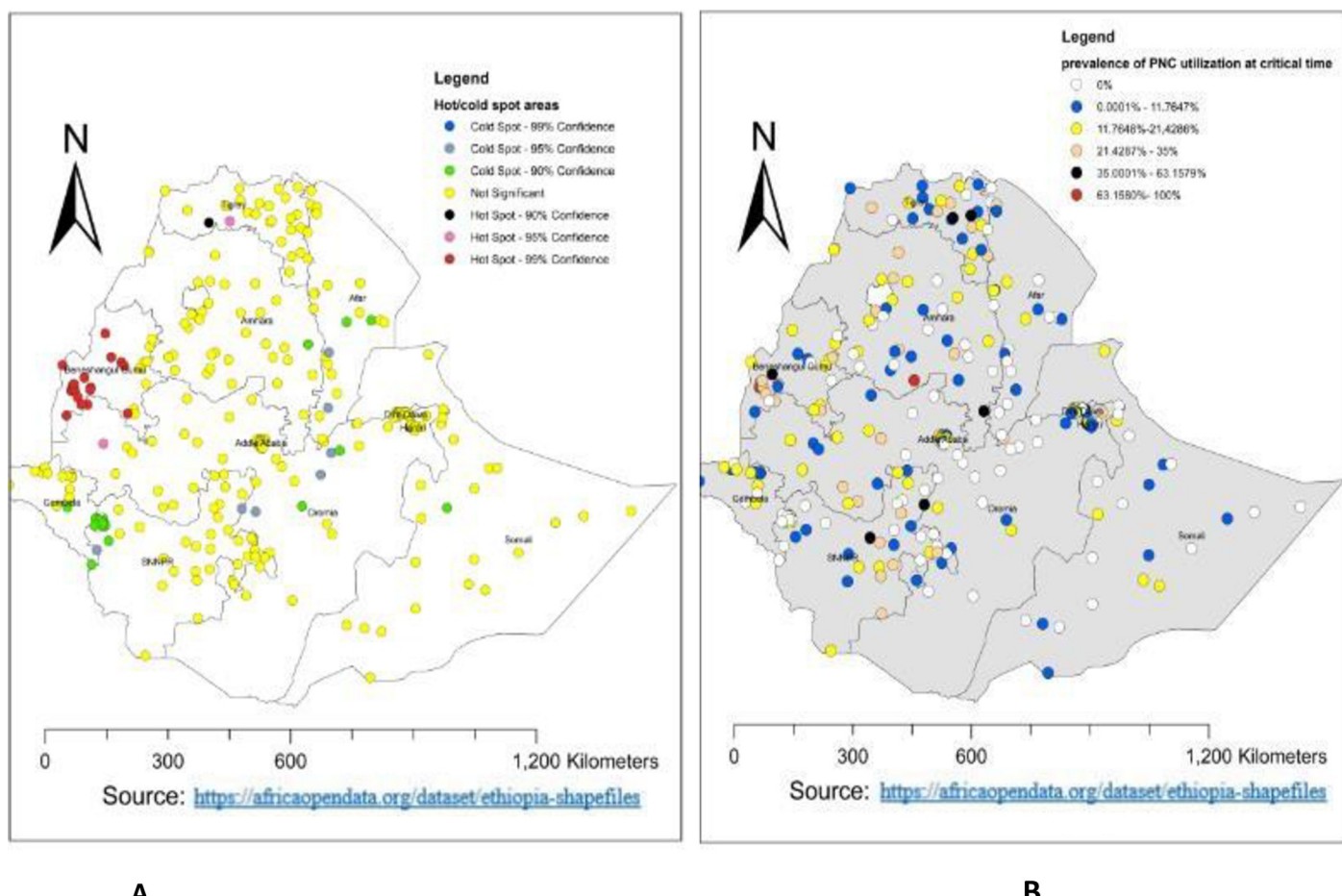

**A**                                                                                                          **B**

**Fig 3. Spatial distribution and hot spot analysis of PNC utilization during the critical time in Ethiopia using EDHS 2019.**

### Factors of PNC utilization during critical time

The variables used in the OLS model explained the model by 56.4%, and the model fulfils all assumptions (17). The overall model's fitness was evaluated using joint Wald statistical significance ($p < 0.001$). The significant predictors were selected by the robust probabilities, which had a p-value of $< 0.01$. Thus, the predictors for PNC utilization during the critical time were the ANC visit, place of delivery, and media exposure (**Tables 4 and 5**).

### Geographic weighted regression

The predictors of PNC utilization during the critical time were not stationary across the clusters in Ethiopia. To consider this non-stationary effect, the local regression method was more advantageous and preferable than the global method [23]. Thus, the adjusted $R^2$ for OLS regression was 56.4%, whereas it was improved to 66.7% in GWR regression. Additionally, the AIC value was reduced from 1301 in OLS to 867.4 in GWR. In general, the local method of analysis (GWR) improves the current model as compared to OLS regression (**Table 5**).

Regarding the predictors of PNC utilization during the critical time, women who had a history of ANC visits had a positive relationship with PNC utilization during the critical time in Ethiopia. Alternatively, as the proportion of women who had a history of ANC visits increased,

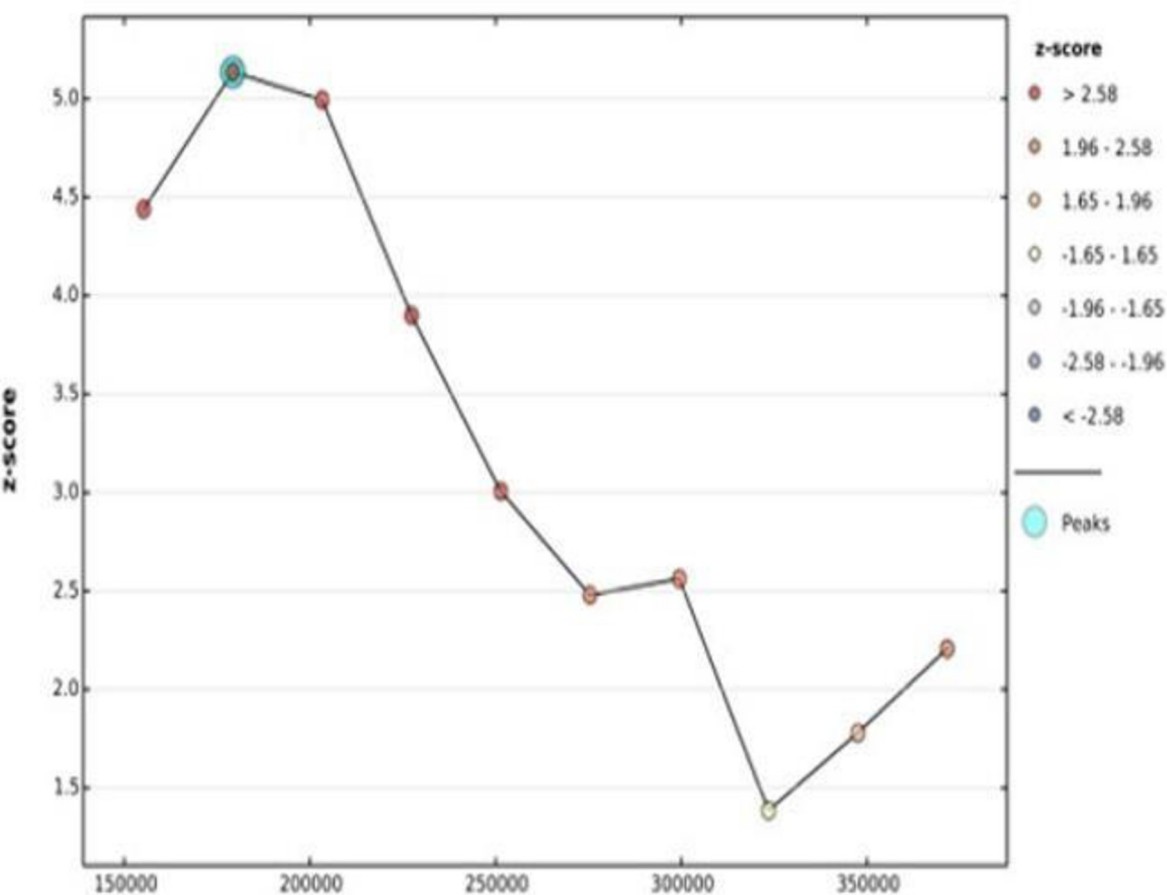

**Fig 4. Incremental autocorrelation of PNC utilization during the critical time in Ethiopia using EDHS 2019.**

the utilization of PNC during the critical time increased as well. The larger the beta coefficient, the stronger the relationship between predictors and outcomes [20]. The red colour in **Fig 6B** indicates ANC visits were the strongest predictors of PNC utilization during the critical time in Benishangul, Gumuz, and Gambela (0.187772–0.23626). But it had a lower effect on the southern nation and nationality of Ethiopia, southern Somalia, and eastern Oromia (0.0069988–0.09861) (**Fig 6B**).

Also, as the proportion of women who had media exposure increased, the utilization of PNC during critical times increased. Geographically, having media exposure had a positive and strong relationship with PNC utilization during the critical time in Diredawa, Harari, the western part of Benishangul Gumuz, and Gambela (0.166868–0.572251). Whereas, a lower positive relationship was observed in Amhara, Tigray, Afar, Addis Ababa, Oromia, and the southern part of the Somalia region (0.0328–0.2498) (**Fig 7A**). Furthermore, as the proportion of women who had no education increased, the utilization of PNC during critical times decreased. Thus, no education had a negative and strong relationship with PNC utilization during the critical time in Benishangul Gumuz, the Amhara region, and western Tigray

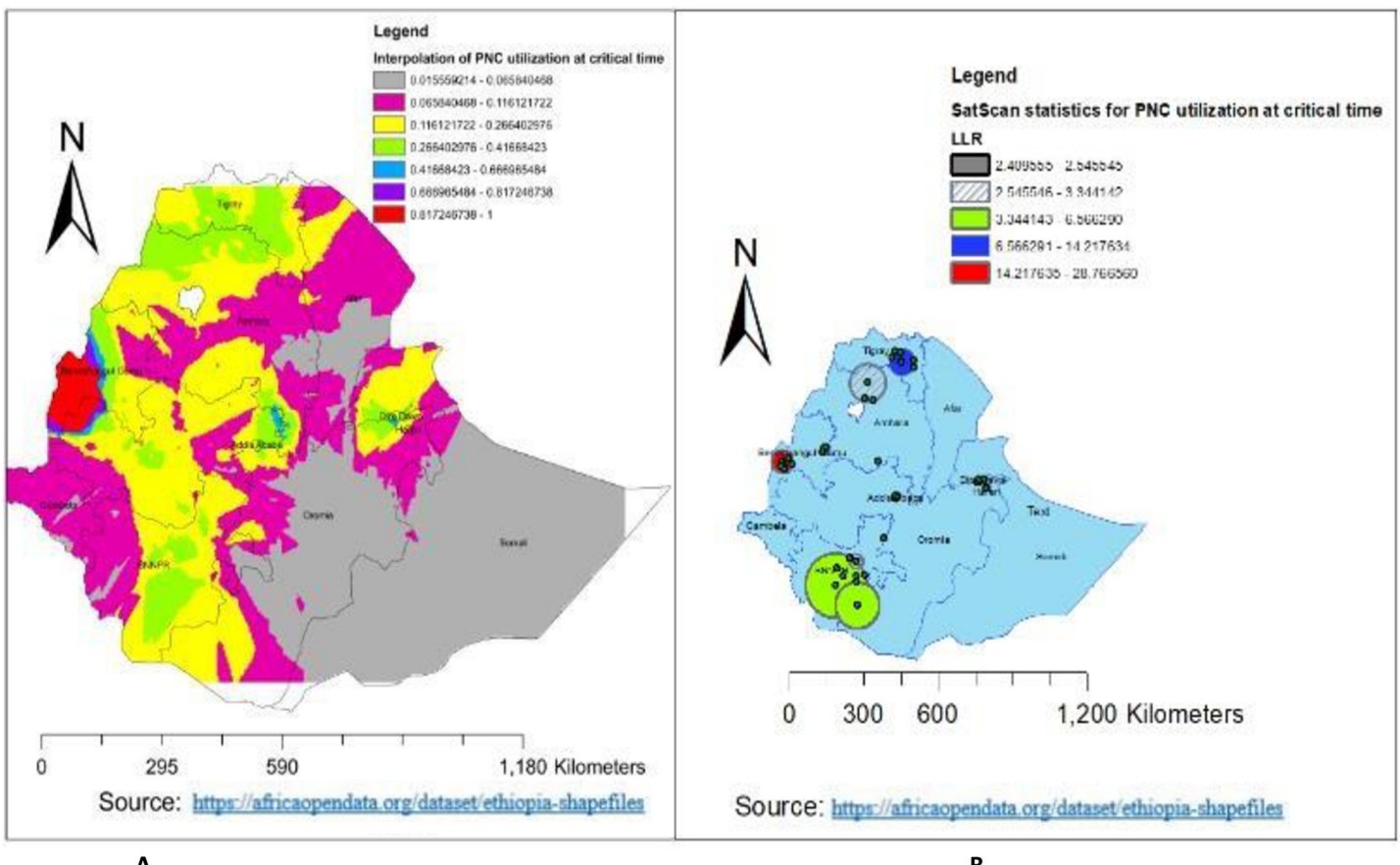

**Fig 5. Spatial interpolation and Sat scan analysis of PNC utilization during the critical time in Ethiopia using EDHS 2019.**

(-0.260170–0.082595). Whereas, a lower negative relationship was observed in Diredawa, northern Tigray, western Oromia, the western part of the south nation and nationality, Oromia, and Somalia (-0.059575–0.034849) **(Fig 7B)**.

## Discussion

PNC utilization during the critical period (within 2 days) is a very vital service for the mother's and the newborn's health. To reverse the complications associated with health problems during the postnatal period, evidence-based, area-specific interventions are very important by assessing the spatial distribution and identifying the predictors in the specific geographical area using geographic weighted regression. Thus, the current study was aimed at assessing the spatial distribution and application of geographical weighted regression analysis to assess the

**Table 3. Significant clusters for PNC utilization during critical time in Ethiopia using 2019 EDHS data.**

| Cluster type | Significant Enumeration Areas (clusters) detected | Coordinate/radius | Population | Cases | RR | LLR | P-value |
|---|---|---|---|---|---|---|---|
| Primary | 157, 151, 153, 147, 152, 149, 154, 150 | (10.196300 N, 34.606731 E) / 41.99 km | 104 | 42 | 3.69 | 28.77 | <0.001 |
| Secondary | 203 | (7.596968 N, 38.357304 E) / 0 km | 19 | 12 | 5.51 | 14.21 | 0.00027 |
| Tertiary | 14, 2, 11, 13, 17, 12, 5 | (13.668519 N, 39.013447 E) / 52.49 km | 88 | 27 | 2.72 | 11.59 | 0.0027 |
| Quarterly | 273, 264, 267, 270, 271, 276, 263, 275, 265, 266, 268 | (8.995815 N, 38.793907 E) / 6.96 km | 90 | 27 | 2.66 | 11.12 | 0.0041 |

**Table 4. Summary of OLS results for PNC utilization during critical time in Ethiopia, using EDHS 2019.**

| Variable | Coefficient | Standard error | t-Statistic | Probability | Robust_SE | Robust_t | Robust_Pr | VIF |
|---|---|---|---|---|---|---|---|---|
| Intercept | 0.26754 | 0.12187 | 1.65987 | 0.07784 | 0.0104778 | 2.7982 | **0.006358** | -------- |
| No education | -0.07023 | 0.03786 | -1.5679 | 0.08350 | 0.035417 | -0.931413 | **0.0015*** | 3.412 |
| Facility delivery | 0.324 | 0.015038 | 2.879 | 0.001 | 0.00554 | 3.76 | **0.00532*** | 1.786 |
| Media exposure | 0.0245 | 0.0349 | 1.5934 | 0.001084 | 0.03675 | 0.07834 | **0.0037*** | 2.963 |
| ANC visit | 0.02682 | 0.014587 | 1.4424 | 0.0180503 | 0.00324 | 2.04833 | **0.0012*** | 3.26 |

predictors of postnatal utilization during the critical time hot spots in Ethiopia using the EDHS 2019 data set. Hence, the prevalence of PNC utilization during the critical time in Ethiopia was 34% (95% CI: 31.5%–36.5%). This finding was higher than the EDHS 2011 (6.4%) and EDHS 2016 reports of Ethiopia (16.3%) [24], Rwanda (12.8%) [8]. and in southern Ethiopia, 24.9% [13]. The possible reason may be because of the increase in awareness, accessibility of the health service, and health-seeking behavior across time. But the current finding was lower than the 2016 demographic survey of Uganda (49.5%) [25] and Nigeria (37%) [26]. This might be associated with the differences in cultural, socioeconomic, and accessibility of the health service across the countries.

The spatial distribution of PNC utilization during the critical time in Ethiopia was non-randomly distributed across the regions and clusters, with a global Moran's index statistics value of 0.027 (p-value = 0.023). Which indicated that the distribution of PNC utilization during the critical time was clustered across the area of Ethiopia. Benishangul Gumuz and the western part of Tigray were the hot spots for PNC utilization during the critical time, and the western part of the southern nation and nationality of Ethiopia were the cold spots. The possible reasons for this variation might be variations in awareness about PNC services, socio-economic differences, variations in access to health services, and security problems across the enumeration area of Ethiopia.

The highly predicted region for the utilization of PNC during the critical period in Ethiopia was Benishangul Gumuz. This might be because access to health services is high in Benishangul Gumuz [27]. In return, it can improve the utilization of PNC services during critical times. Whereas, the lower predicted utilization of PNC during the critical time in Ethiopia was in Somalia, the south-eastern part of Oromia, and the southern Afar region. This may be because

**Table 5. OLS diagnosis and model comparison of PNC utilization during the critical time using EDHS 2019.**

| Numbers of observation | 305 | Akaike's information criteria (AIC) | 1305.39 |
|---|---|---|---|
| Multiple $R^2$ | 56.7 | Adjusted $R^2$ | 55.8 |
| Joint F-statistics | 30.49 | Prob (>F), (5,299) degree | <0.001 |
| Joint wald statistics | 48.26 | Prob (>chi-square), (5) degree of freedom | <0.001 |
| Koenker (BP) statistics | 70.43 | Prob (>chi-square), (5) degree of freedom | <0.001 |
| Jarque-bera statistics | 2.45 | Prob (>chi-square), (2) degree of freedom | 0.127 |
| **Model comparisons** | | | |
| Fitness parameter | OLS model | MGWR model | |
| AICc | 1301 | 867.4 | |
| $R^2$ | 57.5% | 68.8% | |
| Adjusted $R^2$ | 56.4% | 66.7% | |

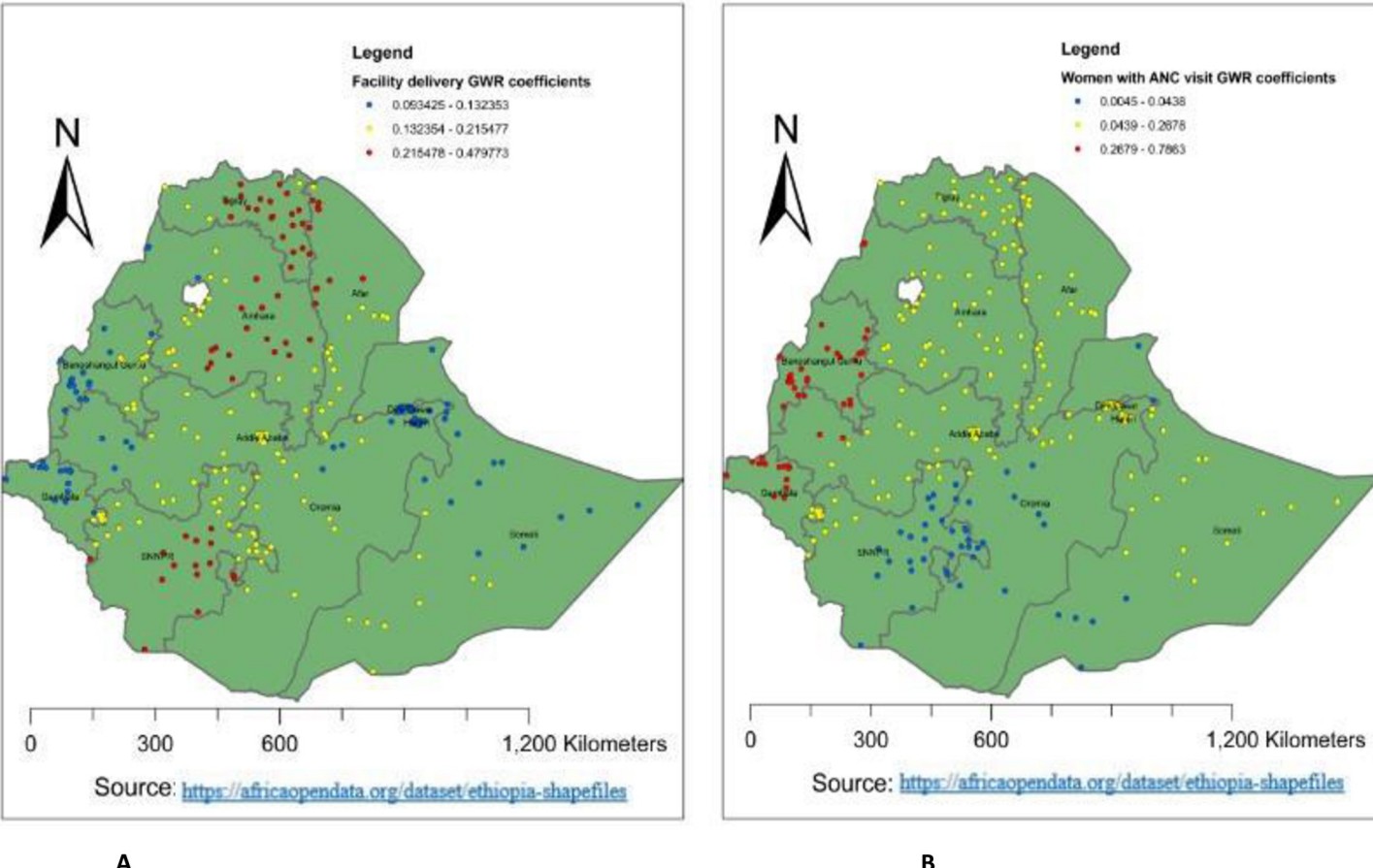

**Fig 6. Women with facility delivery and ANC visit GWR coefficients for predicting PNC utilization during the critical time in Ethiopia, EDHS 2019.**

of the variation in infrastructure, health-seeking behavior, and culture. Additionally, the majority of the Somali Region and Afar Region are pastoralists, and their lifestyle is characterized by seasonal mobility [28]. People in the pastoralist regions have low access to services and infrastructure, live in a traditional setting, and are influenced by cultural and religious values [29]. Because of their cultural and religious practices, pastoralist people have low acceptance of using health services, or PNC utilization during critical times is highly influenced. The primary cluster area of PNC utilization during the critical time in Ethiopia was located at 10.196300 N, 34.606731 E within a 41.99 km radius in Benishangul Gumuz. This is because access to health services in Benishangul Gumuz is high [27], which enables us to improve the utilization of PNC during this critical time. Which was 3.69 times higher than outside the window (RR = 3.69, LLR = 28.77, p-value < 0.001). It was also found that ANC visits, facility delivery, lack of education, and media exposure were the predictors of PNC utilization during the critical time hot spot. As the proportion of women who had a history of ANC visits increased, the utilization of PNC during the critical time increased. Which was supported by a study done in Ethiopia using EDHS 2016 and a study done in southern Ethiopia [4, 30]. This could be because women with ANC visits have an opportunity to get counselling services about PNC utilization. The strong GWR coefficient of women with ANC visits ranges from 0.187772 to 0.23626 in the Benishangul Gumuz, and Gambela. This could be related to high awareness

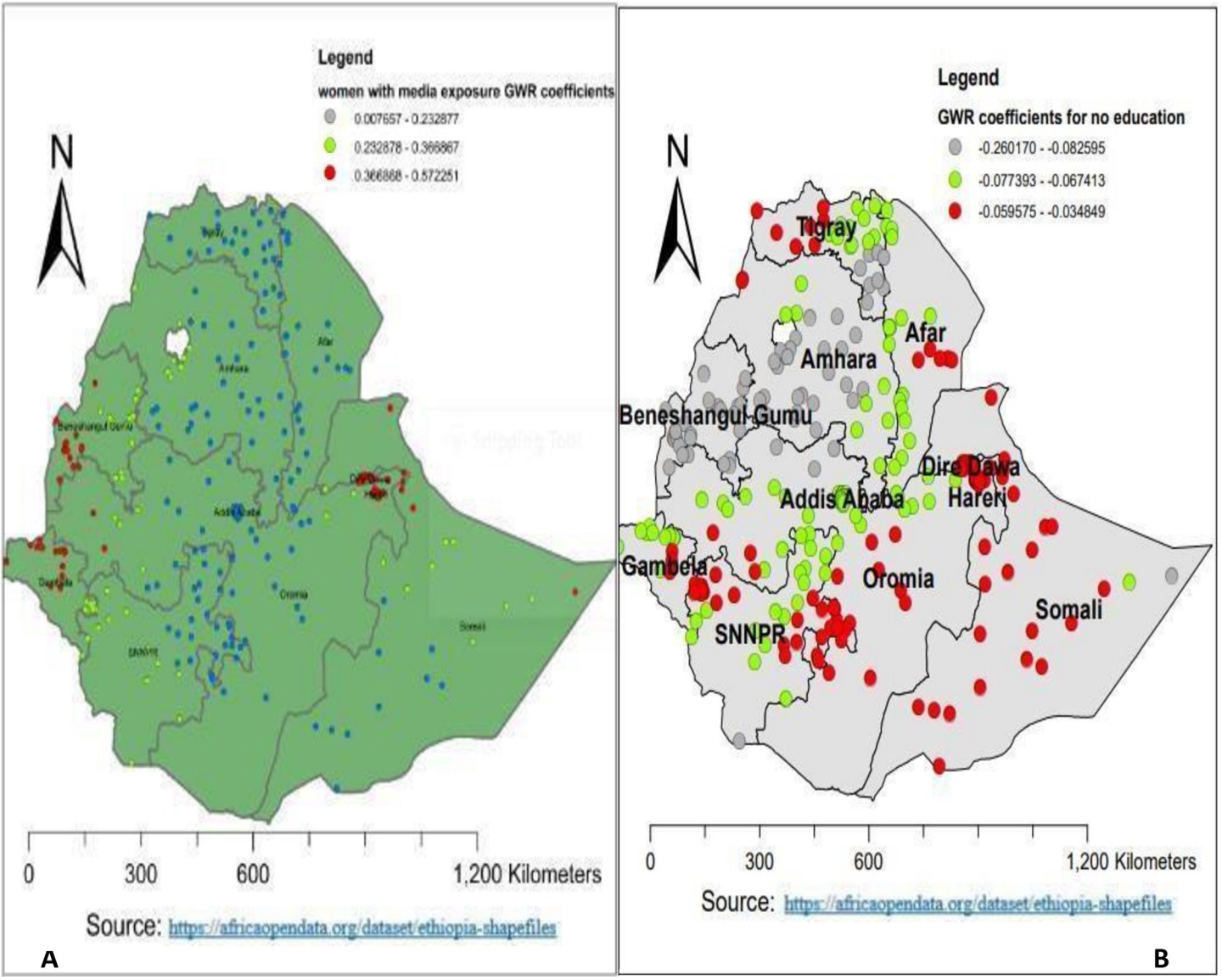

**Fig 7. Women with no education and media access GWR coefficients for predicting PNC utilization during the critical time in Ethiopia, EDHS 2019.**

creation, access to health services, and mobilization activities about PNC utilization during the critical time in these regions [27].

Regarding women with facility delivery, as the proportion of women who deliver at health institutions increased, the utilization of PNC during critical times increased. Which was supported by a study done in Ethiopia using EDHS 2016 [31] and Sidama zone Malga district, southern Ethiopia [13]. This could be due to the fact that women who give birth at the health institution have the chance to access health education and counselling services. The strongest GWR coefficient of facility delivery ranged from 0.215478 to 0.49773 in the eastern part of Amhara, Tigray, and the southern nation and nationality of Ethiopia. This may be associated with the better coverage of delivery service in this area, which provides a good opportunity to counsel PNC utilization during critical times.

Again, as the proportion of women who have media exposure increased, the utilization of PNC during critical time increased. This finding is consistent with another study done in Ethiopia [12], studies in Adwa, Southern Ethiopia, Jabitena Amhara, Kenya and Nepal [32–36]. This may be because media exposure increases the chance of getting information or knowledge about PNC utilization during critical times, and in return, it enables women to utilize this service [37]. Alternatively, media exposure plays a key role in decision-making for women and positively affects health service utilization [38]. In this study, the stronger GWR coefficient of women with media exposure ranges from 0.166868 to 0.572251 in Diredawa, Harari, the western part of Benishangul Gumuz, and Gambela. Whereas, a lower positive relationship was observed in Amhara, Tigray, Afar, Addis Ababa, Oromia and the southern part of Somalia which ranges from 0.0328 to 0.2498. This variability might be related to the difference in media access and habit of attending media across the region of Ethiopia.

Furthermore, as the proportion of women who had no education increased, the utilization of PNC during critical times decreased. A study conducted in Ethiopia, Indonesia, Uganda, and India [12, 39–41] reported the same result. This can be justified by the fact that women with no education lack knowledge about the advantages of early PNC service and have lower health-seeking behavior than women with better educational status. The other possible reason may be that uneducated women can reduce maternal health-seeking behavior and access to the service indirectly through its influence on maternal income. of health services [42]. The stronger negative GWR coefficient of women with media exposure ranges from -0.260170 to -0.082595 in Benishangul Gumuz, the Amhara region, and the western Tigray region. Whereas, a lower negative relationship was observed in Diredawa, northern Tigray, western Oromia, the western part of the south nation and nationality, Oromia, and Somalia, which ranges from -0.059575 to -0.034849. This study provides important information or knowledge for decision-makers or policymakers to expand PNC utilization during critical times by designing key strategies. Additionally, the current study gives important knowledge about the cold spots of PNC utilization during the critical period in Ethiopia, which enables the government and non-governmental organizations to make fair allocations of resources and select the prioritized areas for interventions. The current study identifies the effect of each predictor across the clusters using the most recent, representative, and generalizable sample. But it has limitations, such as the fact that the current study used the geographic coordinates of clusters (2 kilometers for urban areas and 5 kilometers for most clusters in rural areas), which makes it difficult to estimate the cluster effect in the spatial analysis. Secondly, the current study used secondary data, and thus, additional important public health variables were not included.

## Conclusion

In Ethiopia, one-third of women utilize the PNC during critical times. The hot spot area of PNC utilization during the critical time was in Benishangul Gumuz and the western part of Tigray, whereas the western part of the southern nation and nationality of Ethiopia was the cold spot area. Antenatal care visits, facility delivery, lack of education, and media exposure were the predictors of postnatal care utilization during the critical time in the hot-spot areas of Ethiopia. Considering the socio-demographic characteristics, women's empowerment can be a powerful tool for increasing the uptake of early PNC, tracking change over time, and uncovering the needs or strengths of a community to guide planning, policy development, or decision-making. Alternatively, it has the implication of providing an input tool for designing policies and strategies for early PNC utilization. Therefore, in light of the study, the government of Ethiopia should increase access to antenatal care, facility delivery, education, and media to improve the utilization of PNC services during this critical time. Furthermore, future

researchers should conduct research by considering additional clinical and public health-important variables through primary data collection techniques.

## Acknowledgments

The authors would like to give thanks to DHS International for accessing the data.

## Author Contributions

**Conceptualization:** Muluken Chanie Agimas, Nebiyu Mekonnen Derseh, Meron Asmamaw, Habtamu Wagnew Abuhay, Getaneh Awoke Yismaw.

**Data curation:** Muluken Chanie Agimas, Tigabu Kidie Tesfie, Getaneh Awoke Yismaw.

**Formal analysis:** Muluken Chanie Agimas, Tigabu Kidie Tesfie, Nebiyu Mekonnen Derseh, Habtamu Wagnew Abuhay, Getaneh Awoke Yismaw.

**Investigation:** Muluken Chanie Agimas.

**Methodology:** Muluken Chanie Agimas, Meron Asmamaw, Habtamu Wagnew Abuhay, Getaneh Awoke Yismaw.

**Software:** Tigabu Kidie Tesfie, Habtamu Wagnew Abuhay.

**Supervision:** Getaneh Awoke Yismaw.

**Validation:** Meron Asmamaw, Habtamu Wagnew Abuhay, Getaneh Awoke Yismaw.

**Visualization:** Meron Asmamaw, Getaneh Awoke Yismaw.

**Writing – original draft:** Meron Asmamaw, Habtamu Wagnew Abuhay, Getaneh Awoke Yismaw.

**Writing – review & editing:** Meron Asmamaw, Habtamu Wagnew Abuhay, Getaneh Awoke Yismaw.

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
