## [Decision Letter · Decision Letter 0]

20 Feb 2024

PONE-D-23-41347Spatial distribution and application of geographical weighted regression analysis to assess the predictors of postnatal utilization during the critical time hot spots in Ethiopia using EDHS 2019PLOS ONE

Dear Dr. Agimas,

Thank you for submitting your manuscript to PLOS ONE. After careful consideration, we feel that it has merit but does not fully meet PLOS ONE’s publication criteria as it currently stands. Therefore, we invite you to submit a revised version of the manuscript that addresses the points raised during the review process.

We look forward to receiving your revised manuscript.

Kind regards,

Nigusie Selomon Tibebu, MSc

Academic Editor

PLOS ONE

Journal Requirements:

4. Please include your tables as part of your main manuscript and remove the individual files. Please note that supplementary tables (should remain/ be uploaded) as separate "supporting information" files.

Reviewers' comments:

Reviewer's Responses to Questions

**Comments to the Author**

1. Is the manuscript technically sound, and do the data support the conclusions?

Reviewer #1: Partly

Reviewer #2: Partly

2. Has the statistical analysis been performed appropriately and rigorously? 

Reviewer #1: No

Reviewer #2: No

3. Have the authors made all data underlying the findings in their manuscript fully available?

Reviewer #1: Yes

Reviewer #2: No

4. Is the manuscript presented in an intelligible fashion and written in standard English?

Reviewer #1: Yes

Reviewer #2: No

5. Review Comments to the Author

Reviewer #1: Dear editors for PLOS ONE

Thank you for allowing me to review this paper. The paper is addressed important public health topic. I have provided some question and comments as outlined below.

General question

1. Is it really necessary to conduct additional research when a comparable study was completed in Ethiopia?

2. What will it add to the already known facts?

3. What was the gap in the previous study?

4. Why you select geographical weighted regression analysis for this study?

5. Since EMDHS 2019 the data were not collected all variable like the variables; Media exposure such as (internet use, watching TV, listening radio, and reading newspaper), husband education status, maternal working status and so on. These variables were the most significant factor for postnatal utilization during the critical time

General comment

In abstract section particularly in result and conclusion part, it is too lengthy & try to summarize the sentence and it should incorporate the mothed that you used for spatial analysis.

Study design and setting section you should study area map. Since your study, analysis was spatial regression so it is better to show your study area within a single map

In your Sampling procedure and sampling technique section, “2105 women aged 15-49 years who give birth preceding 2019 EDHS survey were included for the study”. It is weighted sample size or not? It should be clear it maybe weighted or unweighted sample size.

In your method section especially spatial analysis part, you should write all spatial analysis technique separately like spatial auto-correlation, hotspot analysis, and interpolation and SaTScan analysis and cite appropriate citation. In addition, why you select ordinary kriging interpolation technique from other interpolation technique like deterministic like inverse distance weighted(IDW) and geostatistical interpolation methods among those methods simple, ordinary Kriging and universal Kriging?

Note: See the results section, especially table 3 for the OLS result. Since your explanatory variables, such as primary education and delivery by CS, were not statically significant variables, your OLS result is unreliable and incorrect. As a result, you should reanalyze your spatial regression by removing those two variables in OLS and GWR analysis and map out all statically significant variables. This breaks the assumption of OLS analysis that violet explanatory variables are statically significant. Rewrite your abstract, results, discussion, and conclusions section overall in light of the updated data.

Reviewer #2: Title: Spatial distribution and application of geographical weighted regression analysis to assess the predictors of postnatal utilization during the critical time hot spots in Ethiopia using EDHS 2019. This article contains some good points but has many limitations and concerns that should be solved before consideration.

Abstract

Introduction: The introduction in the third line states the death rate which does indicate what death. In the next line, it expresses maternal and child mortality. Thus, can you please write the idea sequentially?

The other next statement shows the highest death but not much attention was given. Even considering the research published maternal and child issues account for more than 80% of all articles. If considering several NGOs, government programs, and other aids the issue related to maternal and child health takes the greater portion. Thus, can rewrite the whole abstract in a more plausible scientific way of writing?

Please follow the following more common method of writing an abstract:

The first two lines must encompass the context of the study and the research problem, further two lines must be covered the objective of the papers with unfolding the description of the title. In the next 2 to 4 lines the methodology will be covered. Afterward, the next two lines are for result and performance. In these lines, the author must define how the results and performance are being achieved, for instance, by conducting either simulation or physical implementation. Please mention the name of the simulation or the physical method. The result statistics must be mentioned in the last two lines and either in percentage or with real-time values.- Objective should be clear and precise.

Method: The method mentions the source of data but does not any method of how the data is handled(extracted, missing handled…etc).

The result section even misses the major finding like in proportion or whatever in the very first line.

Conclusion: The conclusion is comprehensive but could be more specific about the optimal PNC and what to do to improve because you just said to improve PNC. Thus, be more practical in both conclusions.

Introduction

- The introduction could be clearer and more concise but still contain more important facts that could clearly show the gap in the literature.

- It provides less extensive background information about PNC and child and its potential impact on maternal health. It might be more effective if the text is more concise and focused on the most crucial information, including clearly identifying the gap in knowledge that this study is addressing. Many studies already discussed this information and it is already overwhelming, if not only for using geographically weighted regression. Reorganizing ideas could be very important and language editing might be also crucial.

-

Method section

The method section could include a brief note on

- Data Preparation

- Brief description of population, sampling, and how the final sampling arrived on

- Description and definition of included variables in table form

- The inclusion and exclusion criteria are well outlined, but the section could be more concise. For easier readability, the extensive list could be presented in a more summarized and organized format, possibly in a table or listed format.

- Brief description of the selected method and its advantages over another method i.e. reason for picking it. Those sections should be under specific subheadings.

- The source of data was mentioned but there is no clear guide other people can access and conduct other studies or there is the link to access the datasets. The link you provided is not the right link. It is the link to the page not to data. just refer to this, it is available in the resource used in this dataset.

- Explanation regarding the possibility of bias and multicollinearity needs addressed

-

Results:

This section look well write, but could benefited from more editing of language, combination of numbers and texts.

Discussion

- The section could provide a more in-depth comparison with existing literature, especially contrasting findings and methodologies. Further, discussing how this study adds to the existing body of knowledge could be more explicit

- The limitations section should be expanded to include limitations related to the methodology used, data, sampling, representatively, inclusion criteria, or analysis methods.

Conclusion:

- The main conclusions of the study should be presented more clearly and concisely. It's essential to summarize the key findings and their implications straightforwardly to ensure that readers can easily grasp the main takeaways from the research.

- The conclusion should include more detailed recommendations for future research, specifically addressing the gaps and limitations identified in the review. Clear and actionable suggestions for future research would enhance the usefulness of the conclusion

- The conclusion could elaborate more on the policy implications of the findings. How should policymakers approach

- Provide more explicit conclusions regarding the importance of considering contextual factors such as socio-demographic characteristics, women’s empowerment, and partner support

6. PLOS authors have the option to publish the peer review history of their article (what does this mean?). If published, this will include your full peer review and any attached files.

Reviewer #1: No

Reviewer #2: No

---

## [Decision Letter · Decision Letter 1]

30 Jul 2024

PONE-D-23-41347R1Spatial distribution and application of geographical weighted regression analysis to assess the predictors of postnatal utilization during the critical time hot spots in Ethiopia using EDHS 2019PLOS ONE

Dear Dr. Agimas,

Thank you for submitting your manuscript to PLOS ONE. After careful consideration, we feel that it has merit but does not fully meet PLOS ONE’s publication criteria as it currently stands. Therefore, we invite you to submit a revised version of the manuscript that addresses the points raised during the review process.

**ACADEMIC EDITOR:**I agree to the comments from the reviewer this paper requires a major revision in terms of conceptual clarity e.g. critical time and language. The author may consider the comments from Reviewer 3 and add a graph/section conceptual framework of the paper. ==============================

We look forward to receiving your revised manuscript.

Kind regards,

Ranjan Kumar Prusty, Ph.D.

Academic Editor

PLOS ONE

Additional Editor Comments:

I agree to the comments from the reviewer this paper requires a major revision in terms of conceptual clarity e.g. critical time and language.

The author may consider the comments from Reviewer 3 and add a graph/section conceptual framework of the paper.

Reviewers' comments:

Reviewer's Responses to Questions

**Comments to the Author**

1. If the authors have adequately addressed your comments raised in a previous round of review and you feel that this manuscript is now acceptable for publication, you may indicate that here to bypass the “Comments to the Author” section, enter your conflict of interest statement in the “Confidential to Editor” section, and submit your "Accept" recommendation.

Reviewer #3: All comments have been addressed

2. Is the manuscript technically sound, and do the data support the conclusions?

Reviewer #3: Partly

3. Has the statistical analysis been performed appropriately and rigorously? 

Reviewer #3: Yes

4. Have the authors made all data underlying the findings in their manuscript fully available?

Reviewer #3: Yes

5. Is the manuscript presented in an intelligible fashion and written in standard English?

Reviewer #3: No

6. Review Comments to the Author

Reviewer #3: (No Response)

7. PLOS authors have the option to publish the peer review history of their article (what does this mean?). If published, this will include your full peer review and any attached files.

Reviewer #3: No

---

## [Editor Report · Decision Letter 2]

21 Aug 2024

Geospatial distribution and predictors of postnatal care utilization during the critical time in Ethiopia using EDHS 2019: A spatial and geographical weighted regression analysis

PONE-D-23-41347R2

Dear Dr. Agimas,

We’re pleased to inform you that your manuscript has been judged scientifically suitable for publication and will be formally accepted for publication once it meets all outstanding technical requirements.

Kind regards,

Ranjan Kumar Prusty, Ph.D.

Academic Editor

PLOS ONE
---

## [Editor Report · Acceptance letter]

1 Oct 2024

PONE-D-23-41347R2 

PLOS ONE

Dear Dr. Agimas, 

I'm pleased to inform you that your manuscript has been deemed suitable for publication in PLOS ONE. Congratulations! Your manuscript is now being handed over to our production team.

Kind regards, 

on behalf of

Dr. Ranjan Kumar Prusty 

Academic Editor

PLOS ONE